# Modeling Patient-Specific CAR-T Cell Dynamics: Multiphasic Kinetics via Phenotypic Differentiation

**DOI:** 10.3390/cancers14225576

**Published:** 2022-11-14

**Authors:** Emanuelle A. Paixão, Luciana R. C. Barros, Artur C. Fassoni, Regina C. Almeida

**Affiliations:** 1Graduate Program, Laboratório Nacional de Computação Científica, Petrópolis 25651-075, Brazil; 2Center for Translational Research in Oncology, Instituto do Câncer do Estado de São Paulo, Hospital das Clínicas da Faculdade de Medicina da Universidade de São Paulo, São Paulo 01246-000, Brazil; 3Institute for Mathematics and Computer Science, Universidade Federal de Itajubá, Itajubá 37500-903, Brazil; 4Computational Modeling Department, Laboratório Nacional de Computação Científica, Petrópolis 25651-075, Brazil

**Keywords:** hematological malignancies, treatment outcomes, CAR-T cell exhaustion, memory pool, functional CAR-T cells, antigen dependent CAR-T expansion

## Abstract

**Simple Summary:**

We present the first mathematical model to describe the multiphasic dynamical treatment response in CAR-T cell kinetics through the differentiation of functional (distributed and effector), memory, and exhausted phenotypes, integrated with the dynamics of cancer cells. The CAR-T cell kinetics are evaluated for various hematological cancers and therapy outcomes, providing insights into promising parameters for long-term therapy investigation.

**Abstract:**

Chimeric Antigen Receptor (CAR)-T cell immunotherapy revolutionized cancer treatment and consists of the genetic modification of T lymphocytes with a CAR gene, aiming to increase their ability to recognize and kill antigen-specific tumor cells. The dynamics of CAR-T cell responses in patients present multiphasic kinetics with distribution, expansion, contraction, and persistence phases. The characteristics and duration of each phase depend on the tumor type, the infused product, and patient-specific characteristics. We present a mathematical model that describes the multiphasic CAR-T cell dynamics resulting from the interplay between CAR-T and tumor cells, considering patient and product heterogeneities. The CAR-T cell population is divided into functional (distributed and effector), memory, and exhausted CAR-T cell phenotypes. The model is able to describe the diversity of CAR-T cell dynamical behaviors in different patients and hematological cancers as well as their therapy outcomes. Our results indicate that the joint assessment of the area under the concentration-time curve in the first 28 days and the corresponding fraction of non-exhausted CAR-T cells may be considered a potential marker to classify therapy responses. Overall, the analysis of different CAR-T cell phenotypes can be a key aspect for a better understanding of the whole CAR-T cell dynamics.

## 1. Introduction

Chimeric antigen receptor (CAR)-T cell immunotherapy is approved by the US Food and Drug Administration to treat several hematological cancers such as B-cell acute lymphoblastic leukemia (ALL), multiple myeloma, mantle cell lymphoma (MCL), diffuse large B cell lymphoma (DLBCL), high-grade B cell lymphoma, primary mediastinal large B cell lymphoma, and follicular lymphoma. Although several malignancies are treated, only two antigens are targeted: CD19 and BCMA (B cell maturation antigen) [1]. All the approved therapies consist of in vitro insertion of a CAR gene on autologous T lymphocytes after the failure of prior therapies, such as chemotherapy or HST transplantation. In a recent review, Huang et al. [2] compared CAR-T therapies approved to date, highlighting their main features.

CAR-T cells promote potent anti-leukemic activity in children and young adults against chemotherapy-resistant B-ALL [3,4]. In a follow-up study, Lee et al. [3] correlated complete response to therapy with greater in vivo expansion of CAR-T cells and recorded the median overall survival of 10.5 months for the entire cohort [4]. Melenhorst et al. [5] reported two cases of patients with chronic lymphocytic leukemia (CLL) who experienced a complete remission response for ten years. These studies highlight the capacity of CAR-T cells to induce a long-term sustained response in patients. On the other hand, Gardner et al. [6] reported the existence of patients with a longer detection of circulating CAR-T cells, but with a constant tumor burden, thus characterizing the presence of non-functional CAR-T cells. This provides evidence of a progressive loss of functionality, expansion capacity, and persistence of the CAR-T cells due to the emergence of the exhausted CAR-T cell phenotype [7].

The success of CAR-T cell immunotherapy mostly depends on the patient’s T lymphocyte characteristics after CAR-T cell manufacturing, such as expansion, persistence, and cytotoxic capacity [8]. Thus, one of the main goals to obtain improved treatment response is avoiding or delaying T cell exhaustion and maintaining the memory phenotype [9]. Exhausted CAR-T cells are known to affect therapy response in pre-clinical studies; to overcome this issue, several strategies on CAR design and in vitro expansion techniques are under investigation [10]. Improvements in CAR design to develop a CAR-T cell exhaustion-resistant phenotype have shown promising results, leading to cell phenotypes with superior antitumor functions and prolonged lifespan [7]. Clinical studies show that there is a correlation between the number of memory cells in the product with a greater CAR-T cell persistence, longer time for disease progression, and sustained remission [11]. Therefore, an analysis of T cell subsets in both the final product and the in vivo samples post infusion may improve the understanding of how immunophenotypes enhance the effectiveness of CAR-T cell therapy [12].

Much effort has been carried out in order to better understand the intrinsic mechanisms that regulate CAR-T immunotherapy, aiming to improve it, expand it to other types of cancers, and increase its success rates. Previous chemotherapy regimens, cytokines used to expand CAR-T cells in vitro, the number of CD3+ cells before apheresis, and even the CAR design (costimulatory domains and affinity to antigen) affect CAR-T cell product and the therapy success [8]. Different responses to therapy profiles are observed in patients: non-response, antigen-negative relapse, antigen-positive relapse, and complete and sustained remission [3,4,13,14]. Dysfunctional CAR-T cells are to blame for non-responses and antigen-positive relapses [15], but the mechanisms associated with loss of functionality deserve more investigation.

In recent years, quantitative models have been developed to depict CAR-T cell kinetics and their interactions with cancer cells. Recent reviews have examined CAR-T structure, physicochemical and pharmacological properties, among other aspects of CAR-T cell immunotherapy [2,12,16,17,18]. They have also analyzed existing quantitative modeling approaches, discussing their usefulness, limitations, and challenges. Unlike traditional drugs, CAR-T cell dynamics do not follow the typical path of absorption, distribution, metabolism/catabolism, and excretion. CAR-T cells are living drugs and display unique cellular kinetic profiles [18,19]. Their characteristics encompass antigen-dependent expansion, heterogeneity of the infused product, and influence of the tumor microenvironment.

It is known that the total number of CAR-T cells circulating in peripheral blood (PB) presents a multiphasic dynamic. The first detailed analysis characterizing in vivo CAR-T cell kinetics across multiple diseases (ALL and CLL) was performed in [20]. In general, after the infusion, the circulating CAR-T cells decline due to the distribution of cells throughout the PB, bone marrow (BM), and other tissues. The distribution period is followed by a rapid expansion until reaching a peak, then followed by a biphasic decline in CAR-T cell numbers. In this way, the in vivo CAR-T cell kinetic profile is basically described by the expansion and persistence phases, with the latter indicating the duration of CAR-T cells in the PB and tissues [20]. The dynamics after the peak were further divided into the contraction and persistence phases in [21]. In a first mixed-effect model, Stein et al. [21] independently modeled these two phases. The contraction phase is described by a fast decline due to the programmed cell death of activated CAR-T cells and the loss of antigen stimulation. The persistence phase presents a slower decline owing to the memory phenotype, which may persist for years or decades. Correlating CAR-T cell kinetics with therapy response profiles and tumor types, Liu et al. [22] found that responding patients exhibit, on average, higher cell expansion rate (and capacity) than non-responding patients, but a lower contraction rate. These two models [21,22] are characterized by the use of empirical piecewise functions and require the explicit definition of the timepoints that separate the phases. Furthermore, with a modeling approach, Mueller et al. [20] investigated the interplay among different phenotypes of CAR-T cells, including naïve, central memory, effector memory, and terminally differentiated effector, and CD19${}^{+}$ tumor cells. Although the estimation of the model parameters was challenging, including issues such as lack of identifiability, the model allowed the identification of factors related to the interindividual variability in the used clinical dataset. By modeling both in vitro and in vivo dynamics of anti-BCMA CAR-T cell therapy, Singh et al. [23] described the multiphasic CAR-T cell kinetics assuming that CAR-T cells are split into effector and memory populations. The authors also investigated CAR affinity and antigen-dependent CAR-T cell expansion in a previous work [24]. Several other biological mechanisms have also been investigated in recent works, including competition between CAR-T cells and normal T cells and the role of stochastic extinction [25], the importance of lymphodepletion before applying CAR-T cell therapy [26], the interplay of proinflammatory cytokines [27], among others. Although these models could represent the total CAR-T cell dynamics, they did not focus on the interplay among different CAR-T cell phenotypes and their relation to multiphasic dynamics.

In previous work, we modeled the dynamics of two CAR-T cell phenotypes, effector and memory CAR-T cells, in a pre-clinical model of hematological cancers [28]. Here, we extend our previous model to describe the CAR-T immunotherapy for hematological cancers in patients, considering different CAR-T cell phenotypes, including engraftment of functional CAR-T cells, memory CAR-T cells, which can be converted to the effector phenotype, and loss of functional capacity when converted to exhausted CAR-T cells. To our knowledge, this is the first model in the literature that considers the multiphasic dynamics of CAR-T cell kinetics through the differentiation of functional, memory, and exhausted phenotypes, integrated with the dynamics of cancer cells in a single model. In vivo CAR-T expansion is considered time-dependent and modulated by the antigen expression. Several individual patients’ characteristics, such as the CAR-T cell expansion capacity and the persistence duration, are incorporated into the model. By fitting the model to patients with different types of hematological cancers and therapy outcomes, we analyze and evaluate the influence of phenotypic differentiation on the dynamics and several kinetic parameters to identify those that best characterize long-term responses.

## 2. Methods

We present a mathematical model to describe the response of CAR-T cell immunotherapy against B-cell malignancies in patients, as schematically described in Figure 1. The model extends the one presented in [28], which was developed for pre-clinical scenarios with immunodeficient mice. Here, the main feature is the multiphasic CAR-T cell dynamics, characterized by (i) the distribution of CAR-T cells in the patient’s body right after infusion, (ii) a quick expansion phase, followed by (iii) a contraction phase of the cell population, and (iv) a long-lasting CAR-T cell persistence. The suggested multiphasic CAR-T cell dynamics are sketched in Figure 1a.

The CAR-T cell population is divided into three phenotypes: functional, memory, and exhausted CAR-T cells. The functional CAR-T cells are activated T lymphocytes that present cytotoxic effects. They are further split into two compartments, distributed and effector cells. This allows the description of the following biological characteristics: (i) circulating CAR-T cells in PB may be detectable only a few days after infusion during which CAR-T cells are distributed in the patient’s body (Lee et al. [3] reported that median values of absolute circulating CAR-T cells are detectable only 5 to 9 days after injection, although the median values of tumor cells in PB have already been reduced by this time); (ii) the duration of the distribution phase depends on the disease (it is very fast in ALLs, being undetectable by blood tests, while it is well marked in DLBCLs [22]); (iii) most of the cells that are infused into the patient do not undergo engraftment (only dozens to hundreds of CAR-T clones give rise to the future in vivo oligoclonal population of CAR-T cells [29]; there are patients for whom it has been reported that only a single clone can be responsible for response and persistence [30,31]).

### 2.1. Mathematical Model

Our model describes the dynamics of an immunotherapeutic treatment with CAR-T cells in patients with hematological cancer. Non-functional CAR-T cells and untranslated T lymphocytes would not engraft and thus, they were not considered in the model. Let $C_{D}$ denote the number of distributed CAR-T cells and $C_{T}$ the number of engrafted CAR-T cells that circulate in the PB and in the tumor niche. These functional CAR-T cells are denoted by $C_{F}=C_{D}+C_{T}$, and consist of CAR-T cells that have cytotoxic activity, quickly recognize specific antigens expressed by the tumor cells, and are ready to target them. Engrafted CAR-T cells may differentiate into memory cells, denoted by $C_{M}$, or become exhausted cells, denoted by $C_{E}$. Our mathematical model for the response of CAR-T cell immunotherapy against B-cell malignancies, denoted by *T*, in patients is given by: (1)$$\frac{dC_{D}}{dt}=-\left(\beta +\eta \right)C_{D}\;,$$



(2)
$$\frac{dC_{T}}{dt}=\eta C_{D}+\kappa \left(t\right)F(T)C_{T}-\left(\xi +\epsilon +\lambda \right)C_{T}+\theta TC_{M}-\alpha TC_{T}\;,$$





(3)
$$\frac{dC_{M}}{dt}=\epsilon C_{T}-\theta TC_{M}-\mu C_{M}\;,$$





(4)
$$\frac{dC_{E}}{dt}=\lambda C_{T}-\delta C_{E}\;,$$



(5)$$\frac{dT}{dt}=rT\left(1-bT\right)-\gamma f(C_{F},T)T\;,$$
in which the total number of functional CAR-T cells is: (6)$$\begin{array}{c} C_{F}=C_{D}+C_{T} \end{array}$$
and the functions κ(t), F(T), and $f(C_{F},T)$ are given by: (7)$$\kappa \left(t\right)=r_{min}+\frac{p_{1}}{1+{\left(p_{2}\;t\right)}^{p_{3}}},\;\;F\left(T\right)=\frac{T}{A+T},$$
(8)$$f\left(C_{F},T\right)=\frac{\frac{C_{F}}{T}}{\vartheta +\frac{a+C_{F}}{T}}.$$

An overview of the considered interaction mechanisms is provided in Figure 1b and the biological meaning and units of each parameter are given in Table 1.

The model assumes that, after infusion, CAR-T cells are distributed in the patient’s body. Distributed CAR-T cells have a natural death rate β, and an engraftment rate η. Since the solution of (Equation 1) is $C_{D}\left(t\right)=C_{D}\left(0\right)e^{-(\beta +\eta )t}$, the total number of engrafted CAR-T cells (EC) can be calculated as: (9)$$\begin{array}{cc} EC & \approx \int_{0}^{\infty }\eta C_{D}\left(t\right)dt=\left(\frac{\eta }{\beta +\eta }\right)C_{D}\left(0\right)\;, \end{array}$$
in which $C_{D}\left(0\right)$ denotes the CAR-T cell dose infused at day 0.

The effector CAR-T cells that engraft correspond to the functional CAR-T cells in the PB and tumor niche. These cells undergo clonal expansion upon contact with tumor cells, which express the target antigen, at a time-dependent patient-specific rate κ(t)F(T). The function F(T) describes the effector CAR-T cell expansion as a process that only occurs in the antigen presence and is limited by intrinsic cell proliferation capacity. This behavior is modeled by the function $0\leq \frac{T}{A+T}\leq 1$ in (Equation 7). The constant *A* defines the number of tumor cells at which the expansion is half of its maximum value. Function κ(t) modulates the antigen-dependent CAR-T cell expansion. At the time of infusion, CAR-T cells are at their maximum expansion capacity, which eventually decays toward a basal level. Such dynamics depend on both patient-specific and product heterogeneity characteristics. The design proposed in (Equation 7) is inspired by the time-dependent expansion rate developed in [32] to model the expansion of memory CAR-T cells. Here, the time-dependent expansion rate κ(t) of effector CAR-T cells depends on four parameters: $r_{min}$ is the basal (or background) CAR-T cell expansion rate [25], $p_{1}$ defines the initial expansion rate, while $p_{2}$ and $p_{3}$ regulate the duration and decay of the expansion rate, respectively. In Figure 1c, we exemplify three different patterns of patient profiles (patients A, B, and C). In these examples, the patients have the same baseline expansion value ($r_{min}$), which is not always the case. Upon the presence of antigen, the maximum rate of CAR-T cell expansion is $p_{1}+r_{min}$. Such expansion is sustained for a certain period, forming a plateau, after which it eventually decays. The plateau width and decay rate are regulated by parameters $p_{2}$ and $p_{3}$. In Figure 1c, patient A displays the highest expansion rate ($\kappa_{max}^{A}$) sustained for a shorter period, followed by a sharp decay. In contrast, patient C has a smaller expansion rate ($\kappa_{max}^{C}$) that lasts for a longer period of time and decays more smoothly, implying a much longer expansion than that shown in patient A. Patient B has an intermediate profile with $\kappa_{max}^{A}>\kappa_{max}^{B}>\kappa_{max}^{C}$. Therefore, the function κ(t) allows modeling different behaviors of in vivo CAR-T cell expansion, taking into account patient-specific characteristics, which ultimately define the strength and duration of each phase in CAR-T cell dynamics.

The effector CAR-T cell population is reduced due to natural death, which occurs at a rate of ξ, and the conversion into memory CAR-T cells, at a rate of ϵ. Over time, effector CAR-T cells lose their proliferative and cytotoxic capacities, becoming exhausted cells at a rate of λ. Note that, once exhausted, these cells die naturally, which is modeled by the term $\delta C_{E}$. Moreover, the effector CAR-T cell population also increases due to the conversion of memory CAR-T cells back to the effector phenotype upon antigen presence. This behavior is modeled by the term $\theta TC_{M}$. In addition, tumor cells inhibit effector CAR-T cells through different mechanisms, which are jointly modeled by the term $\alpha TC_{T}$. This term can model, for example, the inhibitory effect on T cells due to the expression of the intracellular enzyme indoleamine 2,3-dioxygenase (IDO) by the tumor cells [28,33].

The dynamics of memory CAR-T cells are described by the differentiation from effector CAR-T cells, the return to the effector phenotype upon antigen contact, and the natural death at a rate of μ. We assume that the memory phenotype has a longer lifetime than the effector phenotype so that μ<ξ. Note that the memory CAR-T cell phenotype considered here has no cytotoxic activity. In addition, we consider that the formation of the memory pool occurs mainly during the contraction phase of the CAR-T cells [34].

We assume that tumor cells grow following a logistic law, with a growth rate *r* and carrying capacity 1/b. The rate at which functional CAR-T cells kill tumor cells is given by $\gamma f(C_{F},T)$. As suggested in [34,35,36], the saturation function $f(C_{F},T)$ (Equation 8) expresses the access of each functional CAR-T cell to tumor cells. When $T>>C_{F}$, the decrease of tumor cells in time depends exclusively on $C_{F}$, being equal to $\gamma f(C_{F},T)T\approx \gamma \;C_{F}/d$. Otherwise, if $T<<C_{F}$, then the rate of decrease of *T* in time is limited to γT. The term ϑT defines the cell number at which the function reaches half of the maximum saturation.

### 2.2. Experimental Data

To assess the model’s capability to describe real and heterogeneous clinical data, we have selected three data sets from the literature concerning CD19-directed CAR-T cell therapy. Measures of CAR-T cell abundance in PB over time are available for individual patients in [22,37,38]. The first data set refers to pediatric and adult patients with B-ALL, CLL, and DLBCL [22]. All patients were treated with autologous second-generation CAR-T constructs with a 4-1BB (CD137) co-stimulatory domain. Data below the detection threshold were not used for model calibration. For ALL patients, the dataset does not provide information on the administered dose; for these cases, we adopted a value of $1.0\times 10^{8}$ cells, corresponding to the median dose of CAR-positive viable T cells for patients weighing > 50 kg and indicated in tisagenlecleucel package approved by FDA [21]. For this dataset [22], patient outcomes were not considered in the simulation because they were not reported for all patients. We also analyzed data from [38], which consist of CLL patients treated with cells of autologous origin and costimulatory signal provided by the 4-1BB domain. The detection threshold was 25 copies/μg DNA corresponding to $2.5\times 10^{6}$ CAR-T cells. CAR-T cells were administrated with a 3-day split-dose regimen (10%, 30%, and 60%), with the total dose reported for each patient. The last selected dataset was obtained from [37] using the software WebPlotDigitizer [39]. This study covered ALL, CLL, DLBCL, and MCL patients, all of them treated with CAR-T cells with CD28 costimulatory domain and were allogeneic (the T cells were obtained from each recipient’s alloHSCT donor). As the quantification threshold was not informed in [37], we assumed the same value used in [38] since both used quantitative polymerase chain reaction (qPCR) and reported the outcomes of all patients. The individual dose was informed per kg, and we consider that each patient weighed 60 kg. Information on data transformation is detailed in the Appendix A together with the table of the extracted data (Appendix A). The labels used here to identify patients are the same as those defined in the original references.

In [37,38], patient outcomes at the last recorded follow-up were available and model parameters were adjusted accordingly. As in [40,41], we considered that patients with complete response (CR) have clinically undetectable tumor burden (below the detection threshold of $2.5\times 10^{6}$ cells), while partial response (PR) characterizes patients with tumor burden above the detection threshold but below 50% of the initial tumor burden (i.e., $5.0\times 10^{6}$ cells), and stable disease (SD) indicates patients with tumor burden in the range between ±50% of the initial tumor burden (i.e., $5.0\times 10^{6}$ cells < *T* < $1.5\times 10^{7}$ cells). For completeness, patients with tumor burden greater than 50% of the initial tumor burden ($T>1.5\times 10^{7}$ cells) are classified as progressive disease (PD), although no PD patient was analyzed in this work.

### 2.3. Mechanisms Underlying the Multiphasic
Dynamics of CAR-T Cell Therapy

The previously published patient data used here consist of CAR-T cell time courses after injection for different malignancies. It is well known that the dynamical clinical response to CAR-T cell therapy is usually characterized by the following phases: distribution, expansion, contraction, and persistence.

The biological mechanisms underlying this multiphasic response of CAR-T cell therapy can be explained through the analysis of the different time scales occurring in the model dynamics. We first note that the observed clinical data consist of measurements of CAR-T cell counts in the PB or BM samples of patient time courses and do not capture the changes in the composition of the CAR-T population by the different phenotypes. Therefore, we track the behavior of the total CAR-T cell population, which is given by $C=C_{D}+C_{T}+C_{M}+C_{E}$. The total CAR-T cell dynamics are, therefore,
$$\frac{dC}{dt}=-\beta C_{D}+\left(\frac{\kappa (t)T}{A+T}-\xi -\alpha T\right)C_{T}-\mu C_{M}-\delta C_{E}\;.$$

We also refer to the *per capita* rate of change in the total CAR-T cell population, given by: $$\begin{array}{cc} \frac{1}{C}\frac{dC}{dt}= & -\beta \frac{C_{D}}{C}+\left(\frac{\kappa (t)T}{A+T}-\xi -\alpha T\right)\frac{C_{T}}{C}-\mu \frac{C_{M}}{C}-\delta \frac{C_{E}}{C}\;. \end{array}$$

Such expressions, together with considerations about the variation in the composition of the CAR-T cell population, reveal the dominant process at each phase. This systematic analysis allows estimating the model parameters for each patient, as follows. As indicated in the time course data of a general patient depicted in Figure 2a, it is possible to split the dynamics into four well-marked phases, approximated as lines in the log plot of the CAR-T cell population along time, which can be fitted to the data. The slope of each line ($m_{d}$, $m_{e}$, $m_{c}$, and $m_{p}$) can be connected to the corresponding dominant process for modeling the overall patient dynamics. Likewise, the *per capita* population rate of CAR-T cells also helps in assessing whether the slopes agree with the variation presented in each phase of the dynamics (Figure 2b).

*Distribution phase:* at the injection time, the entire CAR-T cell population consists of injected cytotoxic cells that target the tumor cells and may die or engraft. Therefore, in the first phase, the other CAR-T cell populations do not contribute to the model dynamics. Mathematically, we approximate $C_{T},C_{M},C_{E}\approx 0$ and $C\approx C_{D}$. Together with the observation that η<<β, we obtain that the change in the CAR-T cell population is approximated by: $$\frac{dC}{dt}\approx -\beta C_{D},\;\;\mathrm{and}\;\;\frac{1}{C}\frac{dC}{dt}\approx -\beta \;.$$

This shows that the initial decline observed in the CAR-T cell population is due to the loss of injected CAR-T cells that do not engraft. Thus, the decline in the clinical data leads to $m_{d}\approx -\beta$, which provides a first approximation for the parameter β. Figure 2b shows that $m_{d}$ has indeed provided a good approximation for the initial *per capita* rate of the total CAR-T cell population.

*Expansion phase*: the second phase is characterized by an exponential growth of the CAR-T cell population. Assuming that the injected cells already died or engrafted, and that there was not enough time for a substantial part of the engrafted effector population to become exhausted neither transit to a memory phenotype, we have that $C_{D},C_{M},C_{E}\approx 0$ and $C\approx C_{T}$. Let us further assume that the tumor load presents values that still stimulate the CAR-T expansion, i.e., T/(A+T)≈1, and that the tumor immunosuppressive effects are negligible (αT≈0). Finally, assuming that the initial expansion level of the effector CAR-T cells has not decreased to its basal level, we have $\kappa \left(t\right)\approx r_{\min}+p_{1}$. With these assumptions, the dynamics of the CAR-T cell population are approximated by: $$\frac{dC}{dt}\approx \left(r_{\min}+p_{1}-\xi \right)C_{T},\;\;\mathrm{and}\;\;\frac{1}{C}\frac{dC}{dt}\approx r_{\min}+p_{1}-\xi \;.$$

This indicates that the CAR-T cell expansion observed in the second phase results from the net effect of combining the initial expansion of effector CAR-T cells ($p_{1}$), their basal expansion rate ($r_{\min}$), and their natural mortality (ξ). This leads to $m_{e}\approx r_{\min}+p_{1}-\xi$, allowing a first approximation to the values of the parameters $r_{\min}$, $p_{1}$, and ξ. At the end of the calibration process, Figure 2b shows that $m_{e}$ is slightly larger than the *per capita* rate of CAR-T cells during the expansion phase, probably due to the formation of memory and exhausted CAR-T cells.

*Contraction phase*: the third phase is characterized by a steep decline in the CAR-T cell population. During this phase, the expansion rate of effector CAR-T cells reduce to its basal level ($\kappa \left(t\right)\approx r_{\min}$) and the intense decrease of the tumor burden reduce the antigen availability so that κ(t)T/(A+T)≈0. Therefore, the mortality process dominates the dynamics, yielding: $$\frac{dC}{dt}\approx -\xi C_{T}-\mu C_{M}-\delta C_{E}\;.$$

Assuming that the memory CAR-T cells have the lowest death rate and that the memory population is smaller than the others, we have $\mu C_{M}<<\xi C_{T}+\delta C_{E}$. Thus, the dominant process is the mortality of the effector and exhausted CAR-T cells. Further, the exhaustion of the remaining effector cells leads to a higher number of exhausted cells in comparison with effector cells. Therefore, $\xi C_{T}<<\delta C_{E}$. Thus, the strong contraction of the CAR-T cell population after the peak is dominated by the elimination of effector and, mainly, exhausted CAR-T cells: $$\frac{dC}{dt}\approx -\xi C_{T}-\delta C_{E}\approx -\delta C_{E},\;\;\mathrm{and}\;\;\frac{1}{C}\frac{dC}{dt}\approx -\delta \;.$$

This allows for identifying the $m_{c}$ observed in the clinical data with the mortality rate of exhausted CAR-T cells, $m_{c}\approx -\delta$. When comparing such an approximation with that depicted in Figure 2b, observed differences may be owed to the mortality of the other population phenotypes.

*Persistence phase*: finally, the fourth phase is characterized by a slower decline in the CAR-T cell population. Taking into account the previous phases, we may assume that the CAR-T cell population is comprised of nearly only memory cells. Therefore, we have: $$\frac{dC}{dt}\approx -\mu C_{M}\;\;\mathrm{and}\;\;\frac{1}{C}\frac{dC}{dt}\approx -\mu \;.$$

Thus, the slope $m_{p}$ observed in the clinical data resembles the mortality rate of memory CAR-T cells, $m_{p}\approx -\mu$. As shown in Figure 2b, $m_{p}$ actually provides a good approximation for the *per capita* rate of the total CAR-T cell population in the persistence phase.

The present analysis shows that the behavior of each phase is influenced by the specific biological characteristics of the most abundant phenotype. The $C_{D}$, $C_{T}$, $C_{E}$, and $C_{M}$ phenotypes are determinants for the distribution, expansion, contraction, and persistence phases of the in vivo CAR-T cell kinetics, respectively. The identification of the dominant mechanisms underlying each CAR-T cell phase drives the model calibration procedure for each patient-specific scenario, as presented in the Appendix A. Such initial approximations may be over or under estimations depending on the validity of the corresponding assumptions in the individual scenario, which is assessed after the calibration procedure.

### 2.4. Model Settings and Numerical Solution

Model equations (Equations (Equation 1)–(Equation 5)) were solved numerically using the explicit fourth-order Runge–Kutta method [42], with $\Delta t=10^{-4}$ days. The initial tumor burden, not reported for most of the analyzed patients, was always set as T(0) = $10^{7}$ cells. The infused dose of CAR-T cells is the initial condition of the distributed CAR-T cell population ($C_{D}\left(0\right)$). The initial populations of the other phenotypes are assigned to zero since the phenotypic differentiations occur throughout the evolution of the dynamics.

The model was adjusted for each individual patient, considering the corresponding reported data and response to therapy, when available. Some of the model parameters (*r*, *b*, γ, and ϑ) were fixed with values from the literature [26], although the value of γ was also changed for some patients. Extensive tests were carried out to obtain the complete (non-unique) set of parameter values for each patient, whose initial approximations were obtained from the analysis presented in Section 2.3, as described in the Appendix A. All sets of calibrated parameters are shown in Appendix A.

## 3. Results

The proposed model (Equations (Equation 1)–(Equation 5)) is used to describe the multiphasic kinetics of the total CAR-T population for different hematological cancer types and therapy responses. The CAR-T cell population encompasses functional (distributed and effector), memory, and exhausted phenotypes, whose interactions improve the understanding of the key factors that influence the dose–exposure–response relationship. Model calibration was performed for patient-specific data using the multi-step strategy based on the mechanisms underlying the multiphasic dynamics of the CAR-T cell therapy (see Appendix A). Good agreement with the data was then obtained using such a strategy, as shown in Appendix A. The relation between the area under the curve of total CAR-T cells and the corresponding fraction of non-exhausted phenotypes provided a significant characterization of patient responses.

### 3.1. Description of the Cellular Dynamics of CAR-T Therapy Applied to Different Hematological Cancers

We first present the results of fitting the model to the data obtained from Liu et al. [22]. A total of 12 patients were selected and grouped according to the type of cancer: DLBCL, pediatric and adult ALL, and CLL. All patients received a single dose of CAR-T cells on day zero. Patient-specific model simulations of CAR-T cell dynamics are shown in Figure 3, as well as the corresponding time-dependent CAR-T expansion rates (κ(t)). In the Appendix A, we show model simulations on a linear scale of total CAR-T cells and tumor burden in Appendix A and the measures of fit quality in Appendix A.

Figure 3 shows that our model was able to capture the diversity of behavior of CAR-T cell dynamics in different types of cancer and patients. We note that the distribution phase is well marked for patients with DLBCL, as already reported in [22]. Regardless of the duration of the distribution phase, similar goodness of fit is observed for all patients, even in those in which it lasts less, as in pediatric ALL patients (see Appendix A). The total number of engrafted CAR-T cells showed great variation between patients, ranging from two to five-fold magnitude orders lower than the initial dose. Patient L (pediatric ALL) recorded the highest engraftment with EC equal to approximately the infused total CAR-T cell dose (see Appendix A). The expansion phase was also well characterized with CAR-T peak values reached between one and two weeks after the initial dose for most patients. Likewise, the model was also able to represent the dynamics of patient UPN10 whose expansion peak occurred late on day 54. The time of maximum concentration of total CAR-T cells matches the peak of the effector phenotype and marks the beginning of the contraction phase. The exhausted and memory CAR-T cells, which are formed and mostly grow during the expansion phase, peak shortly after the start of the contraction phase. Meanwhile, effector CAR-T cells undergo a sharp decrease and are eventually replaced by exhausted CAR-T cells. The decrease of exhausted CAR-T cells due to natural death eventually leads the CAR-T cell population to be mostly formed by long-lasting memory CAR-T cells, starting in the persistence phase.

The patient-specific expansion profiles described by function κ(t) directly reflect the characteristics of the CAR-T cell dynamics, as shown on the bottom row of Figure 3. Patients with later CAR-T cell peaks have more evident plateaus in their κ(t) functions, whereas there are no plateaus for patients depicting earlier CAR-T cell peaks. Consider, for example, the DLBCL patients. They all have different maximum expansion and decay rates, but the same minimum expansion rate. Their expansion profiles are different: they all exhibit plateaus but with different widths and decays to the minimum value. Patient 30 presented the highest expansion and the latest peak; patient 28 showed less expansion and an earlier peak; and patient 27 exhibited a much slower decay of the expansion rate. However, the therapy performed on patient 28 was able to eliminate the tumor faster and the effector phenotype persisted longer due to the lower natural mortality rate. Pediatric ALL patients, on the other hand, do not have maximal expansion plateaus like those with DLBCL, and the dynamics start with the maximum rate of CAR-T cell expansion which soon decays.

Overall, the CAR-T cell dynamics of the patients analyzed in this section were well represented by the proposed model. The analysis did not take into account the corresponding therapy responses since they were not available for all patients. Nevertheless, it was possible to associate the dynamics of the different phenotypes of CAR-T cells with the observed interindividual variability among the selected patients. In the next section, we analyze a new dataset that integrates the therapy outcomes.

### 3.2. Description of the Cellular Dynamics of CAR-T Therapy Applied to Patients with Different Outcomes

We next analyze the two datasets obtained from [37,38]. To distinguish them, we added the letters B or P before the original identifications of patients selected from Brudno et al. [37] and Porter et al. [38], respectively. Patients from [37] received a single dose of CAR-T cells while fractionated doses (10%, 40%, and 60%) were infused at days 0, 1, and 2 in patients from [38]. The 12 selected patients were then grouped as CR, PR, or SD according to their responses to therapy as assessed by the tumor burden at the last follow-up data. Patient-specific model simulations of CAR-T cell dynamics are shown in Figure 4, as well as the corresponding time-dependent CAR-T expansion rates (κ(t)). In the Appendix A, we show model simulations on a linear scale of total CAR-T cells and tumor burden in Appendix A and the goodness of fit is also shown in Appendix A.

Patients from [37] have less cellular kinetic data than those reported in [38]. For patient B10, for example, there are no data points in the contraction and persistence phases. In cases like this, a good characterization of the behavior of these phases is not possible. Model calibrations in these scenarios were then made in order to fit to the therapy responses. Figure 4 (and Appendix A) shows that the proposed model was able to well represent the diversity of behaviors and responses of the analyzed patients. The total number of engrafted CAR-T cells also showed large variation, ranging from the engraftment of approximately the full dose (Patient P22-PR) to just a single cell (Patients B5-CR and B11-PR). The other patients recorded ECs ranging from the same order of magnitude of the infused CAR-T cell dose to eight orders of magnitude smaller (see Appendix A). The expansion profile of each patient, shown on the bottom row of Figure 4, also reflects the dynamics well: longer and higher plateaus correlate with later peaks and higher CAR-T cell concentrations, although the absolute value of the peak depends on other factors. For example, CAR-T cell peaks reached by patients B15-CR and P1-CR are similar, while $\kappa_{max}^{P1}$ is more than four times greater than $\kappa_{max}^{B15}$. Interestingly, some other interindividual variability emerges from the present analysis, as we show in what follows.

Patient P12 exhibits a peak with a greater abundance of exhausted cells. The population of effector CAR-T cells lasts around 3 weeks due to a sharp decline during the contraction phase. However, there was enough time to form a memory pool, which outnumbers the population of exhausted CAR-T cells for a short period of time. After 72 days, effector CAR-T cells resume growth but at undetectable levels. The memory-effector cell conversion is due to the antigen expressed by cancer cells, which grow undetectable for 6 months when they reach the detection threshold. The CAR-T cell dynamics for Patient P22 were characterized by a very short distribution phase and a higher peak concentration than in patient P12, which allowed greater memory formation. This, combined with a smaller quantity of exhausted cells, delayed the tumor’s relapse, which occurred in 300 days. CAR-T cell persistence with loss of cytotoxic function has already been reported in [6].

Patients who reached SD exhibited good expansion profiles with CAR-T cell peaks formed predominantly by effector cells but with significant exhaustion and undetectable levels of memory. Patients B2 and B10 did not present a detectable persistence phase. However, patient B18 presented this phase mostly formed by the exhausted phenotype.

Among patients in the CR group, all depicted good expansion profiles, with CAR-T cells peaking around day 7 or earlier, except for patients B5 and P2, whose peaks occurred on days 10 and 25, respectively. Patient B12 was the only one that did not exhibit a detectable persistence phase, although their tumor burden remained undetectable for 480 days. Patients P1 and P2 were followed for a long time after therapy. Their expansion rate functions are remarkably different: κ(t) has a narrow initial plateau followed by an abrupt decrease to a relatively high level of basal growth for P1, while the maximum expansion rate of P2 is kept close to the basal level. These properties correlate with the highest expansion and earlier peak for P1 and with the lowest expansion and late peak for P2. With the identification of the distribution and expansion phases of these two patients, it was possible to observe their different engraftment capabilities measured by EC: it was ≈12 cells for patient P1 while P2 reached a much higher value of $\approx 10^{5}$ cells (see Appendix A). Recently, Melenhorst et al. [5] published the 10-year follow-up of these two patients who have remained tumor-free (CR). Then, we extended our model simulations up to 3600 days for these two patients to compare with the long-term additional data provided in [5] (the additional data are shown in Appendix A). Appendix A show the corresponding model predictions that match clinical outcomes, with undetectable tumor burdens up to 10 years. We have also performed a new model calibration for patient P2, taking into account the complete dataset. Parameter values are shown in Appendix A and the model simulation is shown in Appendix A. The main difference between the two simulations for patient P2 is that the persistence phase is shown to last much longer when integrating the additional data from [5].

### 3.3. Assessment of Patient Outcomes through Kinetic Parameters

We next performed additional analyses of the presented simulations in Figure 4 by evaluating other parameters that could be correlated with therapy responses. A typical parameter is an area under the concentration–time curve (AUC), which is calculated here for the fractions of CAR-T cell phenotypes. We computed AUC from day 0 to day 28 (AUC0-28) because therapy response is usually assessed on day 28, but we also evaluated AUC from day 0 to both day 60 (AUC0-60) and day 90 (AUC0-90) to identify whether CAR-T cell dynamics at later times may also play a significant role on the treatment response. The top panel of Figure 5 shows the AUC pie charts for each patient, indicated in each column. The pie radius represents the absolute value of the area. Comparing the areas calculated up to 28, 60, and 90 days, we notice that the pattern of the contribution of each phenotype in the AUC is already defined in the first 28 days of the dynamics in most patients, accompanied by an increase in the quantity of the memory phenotype over time. The CAR-T kinetics of CR patients are characterized by higher fractions of memory phenotype in the AUCs than in SD and PR patients, except in P22. This is probably due to the fact that P22 has undergone Richter transformation and obtained DLBCL arising from CLL with CD19-dim cells at 180 days after infusion [38]. Among the PR patients, P12 had the highest AUCs although with a majority fraction of exhausted cells. SD patients had the lowest AUC values, with significant fractions of exhausted cells and less than 0.01% of memory cells.

The middle panel of Figure 5 shows bar plots for each patient of the total number of CAR-T cells at the time of the peak ($t_{peak}$), denoted by $C(t_{peak})$, with the indication of the quantity of each phenotype, and the ratio between $C(t_{peak})$ and the therapeutic dose ($C(t_{peak})/dose$), which provides the number of times that CAR-T cells expanded with respect to the total infused dose. This ratio may be useful since there are patients with lower doses who exhibit high expansions and the other way around. The results in Figure 5 show that there is no clear relationship between these two parameters and the therapy response. Patients B20 and P2 have quite different expansion profiles but both reached CR; all SD patients display higher $C(t_{peak})/dose$ than patient P1; patient B11 displays an expansion ratio similar to B20, but their AUCs differ in both the absolute values and the fraction of memory cells. For all patients, the CAR-T cells at the peak are made up mostly of functional cells.

Finally, the bottom panel of Figure 5 shows the theoretical (predicted) relapse time ($t_{TR}$) and the number of CAR-T cells at that time ($C(t_{TR})$), with the corresponding quantity of each phenotype. For most patients, the number of CAR-T cells at the time of relapse is very small and below the detection threshold. Patients P12, P22, and B18 exhibited the highest $C(t_{TR})$ values but with practically only exhausted cells. We extended the time of simulations for the decade-long patients P1 and P2 up to 20,000 days. Our predictions indicate a small increase in their tumor burdens at non-detectable levels and thus no theoretical relapse. For completeness, the values used in the construction of Figure 5 are presented in Appendix A.

The previous results have shown that a single parameter was not able to correlate to the therapy response. Regarding AUC0-28, for example, low values were obtained for both CR patients (B15, B12, and P2) and SD patients. Considering only the formation of the memory pool, patient B12, who reached CR, formed fewer memory cells than PR patient P12. Moreover, patients P1-CR and B18-SD have similar profiles of functional CAR-T cells. In an attempt to identify whether any combination of parameters could provide insights into the long-term therapy response, we analyzed the parameters considered in Figure 5. The focus was given on the parameters associated with the beginning of the therapy, encompassing AUC0-28, the corresponding fraction of non-exhausted CAR-T cells ($C_{D}+C_{T}+C_{M}$), and quantities $C(t_{peak})$, $t_{peak}$, and $C(t_{peak})/$dose. The plots with some combinations of two of these parameters at a time for all patient outcomes are shown in Figure 6. The plot between AUC0-28 and the fraction of non-exhausted CAR-T cells, depicted in Figure 6a, yields the best separation among the response groups. All SD patients display low exposure to CAR-T cells (AUC0-28 < $5.0\times 10^{10}$ cell·day) with fractions of non-exhausted cells below 0.8. CR patients have high fractions of non-exhausted CAR-T cells, although their AUC0-28 values vary greatly. PR patients are characterized by intermediate values of AUC0-28. Patient P12 displayed the highest production of exhausted cells, while patients P22 and B11 produced the lowest. However, it is noteworthy that P22 (marked with *▵*) had undergone a CD19 negative relapse and there was no information on the cause of patient B11’s relapse. The separation among the CR, PR, and SD groups, although without clear delimitation, is also observed using the areas up to days 60 and 90. This suggests that the joint assessment of AUC0-28 and the corresponding fraction of non-exhausted CAR-T cells plays an important role in patient-specific long-term outcomes. Other parameter combinations do not provide similar separation among the response groups, as illustrated in Figure 6b,c, which show the plots combining AUC0-28 with $C(t_{peak})$ and $C(t_{peak})/$dose with the fraction of non-exhausted CAR-T cells up to 28 days, respectively. Overall, although the joint assessment of AUC0-28 and the fraction of non-exhausted CAR-T cells seems promising; the present analysis is limited to a small number of patients and must be revised by integrating a larger patient cohort.

## 4. Discussion

We describe CAR-T cell therapy with a multi-compartment model considering the interactions between tumor cells and different CAR-T cell phenotypes: functional CAR-T cells (encompassing infused cytotoxic CAR-T cells and proliferative engrafted cytotoxic CAR-T cells), memory CAR-T cells, and exhausted CAR-T cells. Previous models successfully described CAR-T cell dynamics in a step-wise approach, i.e., by splitting the time domain into intervals corresponding to the distribution, expansion, contraction, and persistence phases, and considering different sets of differential equations in each time interval [21,22,43]. Through a single set of differential equations based on mechanistic hypothesis, our model reproduces the multiphasic dynamics of CAR-T cells over time for different hematological cancers and therapy responses, providing insights into the role of each of the CAR-T phenotypes along the dynamics. By analyzing the underlying mechanisms that govern each phase, we identify the roles of each CAR-T phenotype in each phase: the distribution phase is due to the loss of injected CAR-T cells that do not engraft; the expansion phase is characterized by increased proliferation of engrafted CAR-T cells; the contraction phase results from a decrease in such proliferation together with the exhaustion of effector CAR-T cells; the persistence phase describes the survival and slow decay of memory cells. Besides providing insights into such mechanisms, this analysis also allows us to develop a systematic approach for patient-specific model calibration based on experimental data, which makes this step faster and more accurate.

The division of CAR-T cells into phenotypes and their role in therapy response may also contribute to the identification of possible clinically relevant parameters that correlate with a long-term response. Most of the available data on individual patient response to CAR-T therapy consists of time courses of the total CAR-T cell population. In the literature, there are records of the influence of the maximum quantity of CAR-T cells and the exposure time on therapy outcomes [3,38,44], although relevant parameter markers have not been well established yet. Huang et al. [2] suggested that other factors than dose may help to predict the effectiveness of CAR-T cell therapies due to limited knowledge about dose-response relationships of approved treatments. Furthermore, single evidence on exposure-response relationships may be insufficient to guide the clinical development of optimal therapy. The present work sheds light on the importance of understanding the evolution of CAR-T cell phenotypic constitution for investigating CAR-T therapy responses. Indeed, by analyzing different phenotypic profiles for several hematological cancers and responses, we identified that the joint assessment of AUC0-28 and the corresponding fraction of non-exhausted CAR-T cells during the first 28 days after infusion may play an important role in the therapy outcome. The small number of patients used in this study poses a limitation that must be overcome by considering a larger cohort. Furthermore, from the clinical point of view, the characterization of CAR-T phenotypes in patients still presents challenges due to the very few number of retrieved CAR-T cells. The follow-up samples after treatment are scarce in most cases, making it difficult for the translation of data into the mathematical model. Moreover, there is still a lot of discussion about the exhaustion markers in CAR-T cells [45].

CAR-T cell exhaustion is still not a fully understood factor that may limit the efficacy of the therapy. Exhausted CAR-T cell phenotype is characterized by an in vivo loss of functionality, expansion capacity, and persistence [45]. Thus, detecting the presence of exhausted cells and better understanding their onset and temporal evolution during the therapy can be key to improving results. Techniques to prevent, reverse, or delay the exhaustion process are currently under development [7,9,46]. Beider et al. [45] reported that non-responding patients exhibited a large quantity of exhausted CAR-T cell phenotype. In addition, CAR-T cells from these patients presented functional impairment due to lower in vitro proliferative capacity and interleukin-2 production. Fraietta et al. [11] also reported the greater exhaustion in non-responders, accompanied by a low production of memory CAR-T cells. The results in the present work are in agreement with these findings.

There are cases in the literature reporting that the entire CAR-T cell population observed in a patient after injection derives from a single clone. This suggests that, in some cases, only a fraction of the injected CAR-T cells may engraft, survive, expand, and generate memory [30,31]. Although our model does not explicitly consider the number of clones, we present some cases in which the population of effector CAR-T cells derives from the engraftment of only one or few CAR-T cells. This fact demonstrates the ability of our model to describe different scenarios, including the engraftment of ten to thousands of cells.

The distribution phase presents a variable duration in the different types of hematological cancers. Liu et al. [22] highlighted that patients with DLBCL have a well-marked distribution phase. In their model, the authors considered the existence of the distribution phase only when it is clearly indicated by the experimental data. In contrast, our model assumes that CAR-T cells go through the distribution phase soon after infusion regardless of the type of cancer. Thus, our model is able to represent distribution phases of different durations, from the shortest to the longest (patients with DLBCL). In any case, Huang et al. [2] highlighted the need for a gradual reduction of the sampling frequency, with daily samples over the first days to capture the initial dynamics of CAR-T cells after infusion.

Regardless of the disease, CR patients exhibit diverse patterns of CAR-T cell expansion, ranging from the fastest and most intense to the least intense and most durable [37]. Fraietta et al. [11] reported a greater pool of memory CAR-T cells in CR patients. This memory pool formation pattern was not observed in our simulations. Some CR patients indeed exhibited great formation of memory cells, while others had a very low percentage of this phenotype in the total CAR-T cell composition (see Figure 5). However, we identified that all CR patients exhibit high fractions of non-exhausted CAR-T cells ($C_{D}+C_{T}+C_{M}$). In this way, a variety of dynamics was well represented by the developed model. This is particularly important with regard to the treatment exposure time reflected by the AUC0-28 parameter, which ultimately impacts the long-term therapy outcome. Patients P1 and P2, for example, exhibit high and low AUC0-28 values, respectively, while having low fractions of exhausted cells in the same period. Of note, these two patients represent a clinical milestone, being the first patients treated with CAR-T cell immunotherapy to remain cancer-free for a decade [5,47]. Our model simulations captured those dynamics with the presence of CAR-T cells in the blood during the ten years after infusion. By extending those simulations for an additional ten years, our model predicts that patients P1 and P2 will remain cancer-free for at least another decade.

Assessing the long-term dynamics of each simulation fitted to patient data, we identified that some patients present sustained oscillations of CAR-T cells within the clinically undetectable region (see patient B12 in Appendix A, and patient P2 in Appendix A). This suggests the existence of a stable limit cycle for the ordinary differential equation (ODE) system, leading to period oscillations between CAR-T and tumor cells, always with values below clinically detectable levels. The mathematical analysis of the proposed model is underway in order to identify the conditions for the occurrence and stability properties of the observed cycles. Of note, the cyclic behavior in the range of measurable values is considered not biologically acceptable in [12] because it has never been observed clinically. However, Chong et al. [48] identified two long-term CR patients with CAR-T cells fluctuating between detectable and undetectable levels. Qi et al. [18] stated that this fact may suggest the possible existence of cycles in some patients.

Open-data sharing plays a critical role in the development of clinically useful mathematical models in many areas, including CAR-T immunotherapy. Together with open source software, they impact the potential of model development, bioanalytical methods, and designing methods to better inform the models with refined parameter values. Kast et al. [17] highlighted the challenging task of building models that allow the identification of possible correlations between the various mechanisms present in the dynamics of CAR-T cells after infusion and the clinical responses in scenarios characterized by a lack of data. We perform data collection and curation in different publications, often with limited access and incomplete information. Many publications still do not provide experimental data and information such as clinical quantification threshold, tumor burden after lymphodepletion and before CAR-T therapy, and therapy outcomes per patient. These still existing difficulties, when overcome, can significantly contribute to mathematical modeling and eventually benefit the advancement of cancer biology.

CAR-T cell immunotherapy presents great complexity inherent to the manufactured product and the heterogeneity between patients. We consider this variability in the model through the κ(t) function. It directly describes patient-specific characteristics that interfere with the capacity and duration of expansion, and in the basal proliferation level of CAR-T cells. The heterogeneity of the infused product is also contemplated by this function implicitly. Note that our study involved different designs (CAR-T19 BBζ and CAR-T19 28ζ) and origins (autologous and allogeneic) of CAR-T cells as well as different dosing regimens (single and fractionated doses). The κ(t) function was able to handle all the variability existing in these factors. As discussed by Liu et al. [22], these factors together with the product heterogeneity, patient responses, and disease types, also considered in the scenarios assessed here, may generate substantial variability in the kinetics of CAR-T cells. Therefore, in the results presented here, we have observed a great variability of the profiles of the κ(t) function, allowing us to describe the diversity of patient-specific patterns of antigen-dependent expansion. The review work presented in [2] identified some limitations of current mathematical models for describing adoptive cellular therapies in general. They mainly encompass not considering T cell heterogeneity (different phenotypes, subsets, and levels of exhaustiveness) and the role of tumor cells (antigen-expressing cells) in the different phases of T cell kinetics. With the focus on CAR-T cell therapy, Chaudhury et al. [12] also emphasized the importance of considering the phenotypic differentiation of both the infused product and those that eventually occur in vivo in the modeling. The model proposed here is able to circumvent those limitations. Qi et al. [18] emphasized the importance of developing models to describe the kinetics of CAR-T cells and their possible contributions to the identification of key parameters and their relationship with response to therapy. Overall, we hope that the proposed model may contribute to this development, generating insights into many challenges that still remain in CAR-T cell immunotherapy.

## 5. Conclusions

The kinetic profiles of the CAR-T cell population in patients exhibit a great variability among patients and among diseases. The proposed multi-compartment model encompassing CAR-T subpopulations of functional, memory, and exhausted cells was able to capture the dynamics of the therapy regardless of such variability. Further, the observation of some parameters of the CAR-T cell phenotypic dynamics and therapy outcomes suggest that the joint assessment of the AUC0-28 and the corresponding fraction of non-exhausted CAR-T cells has the potential to be used as a predictive marker of the long-term response to CAR-T immunotherapy. The expansion of the study presented here involving a larger cohort will enhance the understanding of the variability in patient-specific treatment responses.

## Figures and Tables

**Figure 1 cancers-14-05576-f001:**
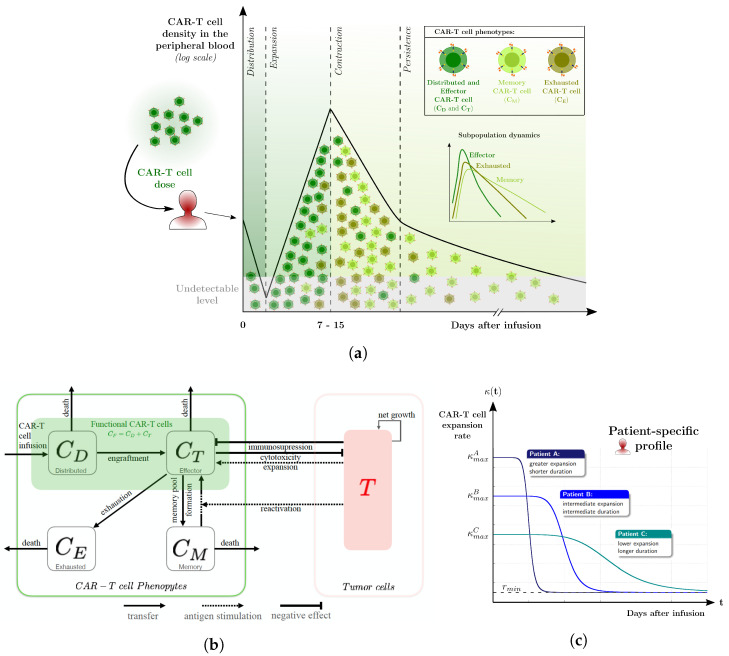
Multiphasic CAR-T cell dynamics. (**a**) Schematic representation of the multiphasic response to CAR-T cell therapy as described by the model. After CAR-T infusion, the typical response profile shows a rapid and usually undetectable decline of circulating CAR-T cells, featuring the distribution phase. The expansion phase is characterized by the antigen-mediated proliferation of engrafted CAR-T cells and reaches its peak about two weeks after starting treatment. Over time, effector cells contribute to the formation of the memory cells but also lose proliferative and cytotoxic capacities, becoming exhausted CAR-T cells. The expansion peak is followed by a contraction phase, marked by a sharp decay of CAR-T cells. The longer lifespan of memory cells leads to a long period of smooth decline, which characterizes the persistence phase. (**b**) Schematic description of the CAR-T immunotherapy model in patients. The CAR-T cells infused into the patient undergo a rapid distribution phase ($C_{D}$). Part of these cells undergoes engraftment and settles in the blood and tumor niche. Engrafted cells are called effector CAR-T cells ($C_{T}$) and expand upon antigen contact, differentiate into memory CAR-T cells ($C_{M}$), become exhausted ($C_{E}$), and die naturally or are targeted by tumor immunosuppressive mechanisms. Both $C_{T}$ and $C_{D}$ populations present cytotoxic effects and, therefore, are named functional CAR-T cells. Memory CAR-T cells die naturally but are readily responsive to antigen-positive cells. When they interact with tumor cells, they differentiate back into effector CAR-T cells, producing a rapid immune response against the tumor. Over time, effector CAR-T cells become exhausted and are eliminated. Tumor cells (*T*), which express the specific target antigen, grow depending on the resources available in the microenvironment and are killed by functional CAR-T cells. The net growth of tumor cells results from the balance between their proliferation and natural death. (**c**) Illustrative patient profiles of the function κ(t), describing the antigen-mediated and time-dependent expansion rate of engrafted CAR-T cells. Individual characteristics and heterogeneity of the infused product ultimately define the strength and duration of each phase of the CAR-T cell dynamics.

**Figure 2 cancers-14-05576-f002:**
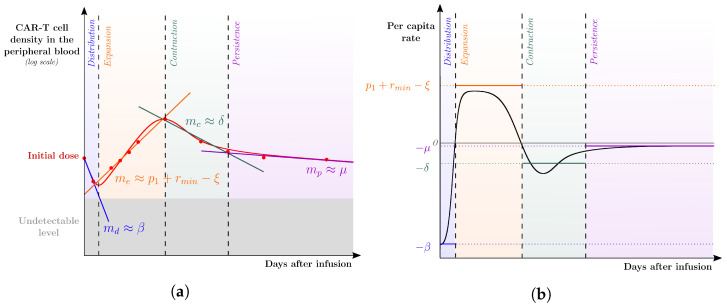
Schematic description of the strategy used to assess the dominant mechanisms underlying the multiphasic CAR-T cell dynamics. (**a**) Experimental data (
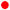
) of CAR-T cell kinetics from a representative patient profile were split among the four phases of CAR-T cell dynamics to which lines in the logplot of the CAR-T cell population along time were fitted. The corresponding (growth or decline) rates are denoted by $m_{d},m_{e},m_{c}$, and $m_{p}$, associated with the distribution, expansion, contraction, and persistence phases, respectively. These rates are used as first approximations to the parameters of the leading mechanism(s) of each phase. Specifically, the distribution phase is mainly driven by the reduction rate of the injected CAR-T cells so that $m_{d}\approx \beta$; the expansion phase is driven by the combined effect between the $C_{T}$ expansion ($p_{1}+r_{min}$) and mortality (ξ), leading to $m_{e}\approx p_{1}+r_{min}-\xi$; the contraction and persistence phases are mainly driven by the mortality of exhausted and memory CAR-T cells, respectively, which yield $m_{c}\approx \delta$ and $m_{p}\approx \mu$. (**b**) The *per capita* rate of the total CAR-T cell population ($C_{D}+C_{T}+C_{E}+C_{M}$) is displayed over time after infusion together with the calibrated values of β, $p_{1}+r_{min}-\xi$, δ, and μ.

**Figure 3 cancers-14-05576-f003:**
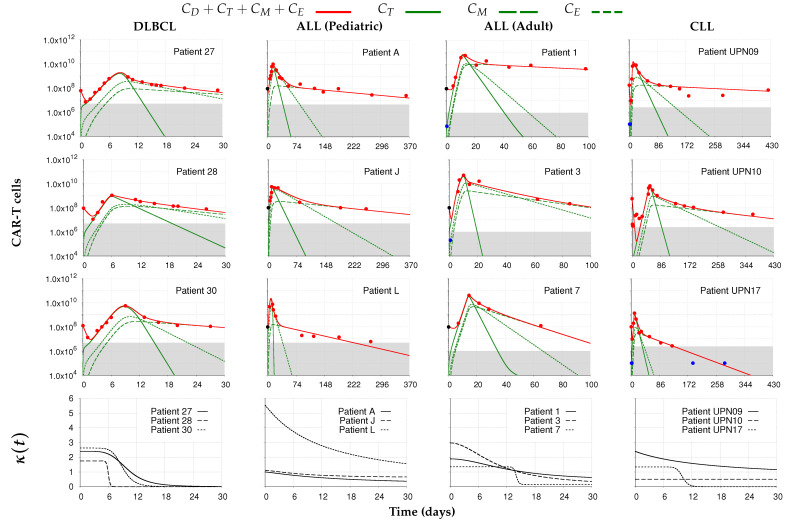
Model simulations fitted to the experimental data (
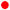
) from [22]. Each column corresponds to the dynamics of the total CAR-T cell population (

) for different diseases (DLBCL, pediatric and adult ALL, and CLL) and different patients. The total CAR-T cell population is divided into effector ($C_{T}$), memory ($C_{M}$), and exhausted ($C_{E}$) phenotypes, shown in continuous, dashed, and dotted green, respectively. The mean dose value of $1.0\times 10^{8}$ cells (
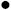
) presented in [21] is used as a surrogate for the actual doses when not reported for patients with ALL. The gray region represents the undetectable levels (below the threshold of $5.0\times 10^{6}$ cells to DLBCL and pediatric ALL, $1.0\times 10^{6}$ cells to adult ALL, and $2.5\times 10^{6}$ cells to CLL [22]). Data points in this region (
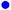
) were not used for calibration and error calculation due to their high uncertainties. The bottom row presents the time-dependent expansion rate function (κ(t)) for each patient.

**Figure 4 cancers-14-05576-f004:**
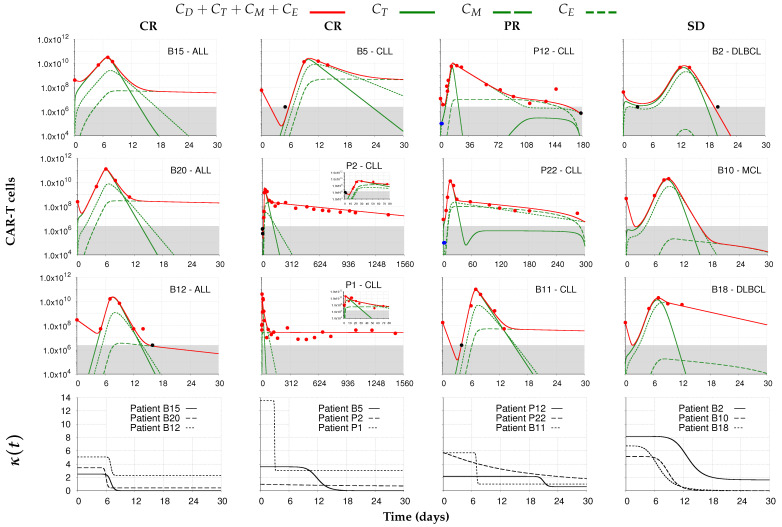
Model simulations fitted to the experimental data (
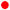
) from [37,38]. Each column corresponds to the dynamics of the total CAR-T cell population (

) for different therapy responses at the last follow-up (interval from infusion to the last follow-up in days) (CR—complete response, PR—partial response, and SD—stable disease) and different patients. The total CAR-T cell population is divided into effector ($C_{T}$), memory ($C_{M}$), and exhausted ($C_{E}$) phenotypes, shown in continuous, dashed, and dotted green, respectively. The gray region represents the undetectable level. Data points (
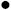
) may assume any value in this region, but some (
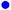
) were not used for calibration and error calculation of the model due to their greater uncertainty. The bottom row presents the time-dependent expansion rate function (κ(t)) for each patient.

**Figure 5 cancers-14-05576-f005:**
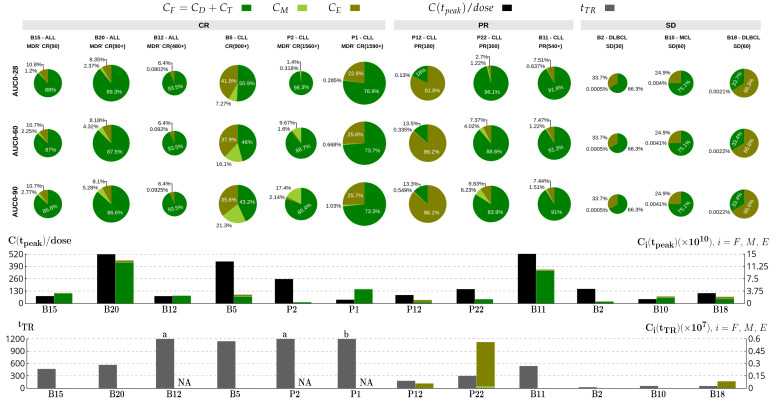
CAR-T therapy outcomes for different patients (one in each column) with ALL, CLL, MCL, or DLBCL with different therapy responses (CR—complete response, PR—partial response, and SD—stable disease). Pie charts display the fractions of functional, memory, and exhausted cells composing the area under the curve of the entire CAR-T population during the first 28 (first row), 60 (second row), and 90 (third row) days after infusion; absolute values of AUC are indicated by the radii of the pies. The bar plot in the fourth row shows the ratio between the CAR-T cell peak value and the infused dose (in black), along with the quantity of each CAR-T cell subpopulation (green tones) on the day when the subpopulation is at the peak time. The bar plot at the bottom presents the time of the theoretical relapse ($t_{TR}$, gray) and the corresponding CAR-T cell subpopulations (green tones). NA (not applicable) indicates no theoretical relapse due to: (a) the occurrence of the limit cycle at undetectable levels; (b) no tumor recurrence within 20,000 days.

**Figure 6 cancers-14-05576-f006:**
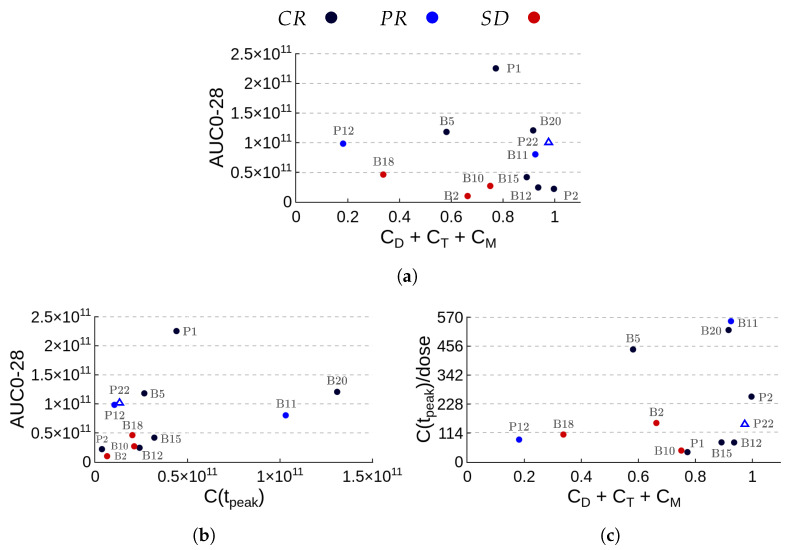
Summary of patient outcomes with respect to kinetic parameters in appropriate units. Panel (**a**) highlights the importance of the fraction of non-exhausted cells ($C_{D}+C_{T}+C_{M}$) in the first 28 days after infusion together with AUC0-28. Patient outcomes are concentrated in three distinct regions. Fully responsive patients (CR) have high fractions of non-exhausted cells while patients who achieved stable disease (SD) have small AUC0-28 values with fractions of non-exhausted CAR-T cells below 0.8. Of the three patients who achieved partial response (PR), patient P12 displays a moderate value of AUC and a high percentage of exhausted cells. Patients P22 and B11 also have moderate AUC values but a small fraction of exhausted cells. Of note, there is no information on the cause of patient B11 relapse but patient P22 (marked with *▵*) underwent mutation with CD19-dim cells. The relationships between AUC0-28 and $C(t_{peak})$ (**b**), and $C(t_{peak})$/dose and fractions of $C_{D}+C_{T}+C_{M}$ (**c**) are also considered but do not provide clear separation among the response groups.

**Table 1 cancers-14-05576-t001:** Model parameters with their units and biological meanings.

Parameter	Unit	Biological Meaning
β	day${}^{-1}$	Reduction rate of infused cells due to natural death during their distribution in the patient’s body
η	day${}^{-1}$	Engraftment rate of injected cells to blood and tumor niche
$r_{min}$	day${}^{-1}$	Minimum expansion rate of effector CAR-T cells
$p_{1}$	day${}^{-1}$	Initial expansion rate of effector CAR-T cells
$p_{2}$	day${}^{-1}$	Rate that regulates the duration of maximum expansion period of effector CAR-T cells
$p_{3}$	-	Expansion coefficient that regulates the decay of maximum expansion period of effector CAR-T cells
*A*, *a*	cell	Half-saturation constants of functions F(T) and $f(C_{T},T)$
ξ	day${}^{-1}$	Death rate of effector CAR-T cells
ϵ	day${}^{-1}$	Conversion rate of effector CAR-T cells into memory CAR-T cells
λ	day${}^{-1}$	Exhaustion rate of effector CAR-T cells
θ	(cell.day)${}^{-1}$	Conversion coefficient of memory CAR-T cells into effector CAR-T cells due to interaction with tumor cells
α	(cell.day)${}^{-1}$	Inhibition coefficient of effector CAR-T cells due to interaction with tumor cells
μ	day${}^{-1}$	Death rate of memory CAR-T cells
δ	day${}^{-1}$	Death rate of exhausted CAR-T cells
*r*	day${}^{-1}$	Maximum growth rate of tumor cells
*b*	cell${}^{-1}$	Inverse of the carrying capacity of tumor cells
γ	day${}^{-1}$	Cytotoxic rate of the functional CAR-T cells on tumor cells
ϑ	-	Half-saturation constant of the cytotoxic effect on tumor cells

## Data Availability

Source data are provided in this paper and all data used in this study can be downloaded from the cited sources. All information, such as parameter values, to replicate simulations and analysis are available in this work.

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
