# Peer review of "Modeling Patient-Specific CAR-T Cell Dynamics: Multiphasic Kinetics via Phenotypic Differentiation"

_cancers, 2022, doi:10.3390/cancers14225576_

Round 1

Reviewer 1 Report

The authors use a mathematical model together with patient samples to analyse the different CAR-T cell phenotypes and their dynamics after infusion. 

The paper is well-written, and provides a way forward for using such samples and models in further research with the goal of understanding even more of the phenotype dynamics impact on long-term responses.

Except that there are some orthographical errors here and there in the text, I have no further comments. 

Author Response

The authors would like to thank the reviewer for the observations. We have performed a thorough English revision to correct the orthographical and grammatical errors found in the text. Modifications to the original manuscript are made in blue. We have also corrected and revised the Supplementary Material.

Reviewer 2 Report

The authors are presenting an interesting article on the modelisation of cart-t cell dynamics taking into account the cart-T cell phenotype and the cancer cells "activity". The objective is clear, the results presention seems accurate and the discussion is highly dynamic. I would have only few questions to help, in my point of view, the readers:

- the authors based their model validation on three data sets. It would have been interesting to give  brief details about the analysis process instead of letting the reader to take a deep look on the litterature. 

- A discussion should be done on the interest of following CART-T cells with molecular imaging, since it could be easier to have an overall view of the situation.

-The injected dose is missing indeed from the 3 data sets. Would have not be possible to contact the corresponding authors to obtain such crucial information for the study. 

Author Response

The authors would like to thank the reviewer for the raised questions. We remark that:

- In the manuscript (Section 2.2 – Experimental data), we provided a description of each dataset with the most relevant information regarding the present work. This description includes the cancer type, available experimental measurements, details about the CAR-T construction, dose information, and patient outcomes.

- To include additional data in our methodology, including molecular imaging, seems very promising. We hope to consider this feature in future work.

- Information on the infused dose is indeed crucial, although it has not compromised the development and analysis of the presented work. As suggested by the reviewer, we will contact the corresponding authors of the mentioned study to obtain the dose information to be used in future work. Of note, an extension of the current work to a larger cohort of patients is underway and the acquisition of such data would certainly play an important role in the analysis.

Of note, we have performed a thorough English revision to correct the orthographical and grammatical errors found in the text. Modifications to the original manuscript are made in blue.

Reviewer 3 Report

The manuscript can be accepted for publication. 

The authors presented a mathematical model which described the multiphasic chimeric antigen receptor (CAR)-T cell dynamics resulting from the interplay between CAR-T and tumour cells, considering patient and product heterogeneities. The topic presented in this manuscript is original and relevant in the field of cancer. The analysis of different CAR-T cell phenotypes can be a crucial aspect for a better understanding of the whole CAR-T cell dynamics. The authors presented the first mathematical model to describe the multiphasic dynamical treatment response in CAR-T cell kinetics through the differentiation of functional (distributed and effector), memory, and exhausted phenotypes, integrated with the dynamics of cancer cells. The methodology discussed is appropriate.  Conclusions are consistent with the results presented and address the main question of the research proposed. References are appropriate and recent literature is cited in the manuscript. The tables and figures are appropriate and are of good quality.

Author Response

The authors would like to thank the reviewer for the encouraging remarks. Of note, we have performed a thorough English revision to correct the orthographical and grammatical errors found in the text. Modifications to the original manuscript are made in blue.